# Empirical Mode Decomposition-Based Filter Applied to Multifocal Electroretinograms in Multiple Sclerosis Diagnosis

**DOI:** 10.3390/s20010007

**Published:** 2019-12-18

**Authors:** Luis de Santiago, M. Ortiz del Castillo, Elena Garcia-Martin, María Jesús Rodrigo, Eva M. Sánchez Morla, Carlo Cavaliere, Beatriz Cordón, Juan Manuel Miguel, Almudena López, Luciano Boquete

**Affiliations:** 1Biomedical Engineering Group, Department of Electronics, University of Alcala, 28801 Alcala de Henares, Spain; luis.desantiago@uah.es (L.d.S.); carlo.cavaliere@uah.es (C.C.); jmanuel.miguel@uah.es (J.M.M.); almudena.lopez@uah.es (A.L.); 2School of Physics, University of Melbourne, Melbourne, VIC 3010, Australia; miguel.ortizdelcastillo@unimelb.edu.au; 3Department of Ophthalmology, Miguel Servet University Hospital, 50009 Zaragoza, Spain; egmvivax@yahoo.com (E.G.-M.); beatrizcordonc@gmail.com (B.C.); 4Aragon Institute for Health Research (IIS Aragon), Miguel Servet Ophthalmology Innovation and Research Group (GIMSO), University of Zaragoza, 50009 Zaragoza, Spain; 5RETICS-Oftared: Thematic Networks for Co-operative Research in Health for Ocular Diseases, 28040 Madrid, Spain; 6Department of Psychiatry, Research Institute Hospital 12 de Octubre (i+12), 28041 Madrid, Spain; emsmorla@gmail.com; 7Faculty of Medicine, Complutense University of Madrid, 28040 Madrid, Spain; 8CIBERSAM: Biomedical Research Networking Centre in Mental Health, 28029 Madrid, Spain

**Keywords:** multiple sclerosis, multifocal electroretinogram, empirical mode decomposition, electrophysiology, biomarker

## Abstract

As multiple sclerosis (MS) usually affects the visual pathway, visual electrophysiological tests can be used to diagnose it. The objective of this paper is to research methods for processing multifocal electroretinogram (mfERG) recordings to improve the capacity to diagnose MS. MfERG recordings from 15 early-stage MS patients without a history of optic neuritis and from 6 control subjects were examined. A normative database was built from the control subject signals. The mfERG recordings were filtered using empirical mode decomposition (EMD). The correlation with the signals in a normative database was used as the classification feature. Using EMD-based filtering and performance correlation, the mean area under the curve (AUC) value was 0.90. The greatest discriminant capacity was obtained in ring 4 and in the inferior nasal quadrant (AUC values of 0.96 and 0.94, respectively). Our results suggest that the combination of filtering mfERG recordings using EMD and calculating the correlation with a normative database would make mfERG waveform analysis applicable to assessment of multiple sclerosis in early-stage patients.

## 1. Introduction

Multiple sclerosis (MS) is a neuroinflammatory disease that demyelinates and degenerates the central nervous system (CNS). It affects over 2 million people worldwide, mainly young adults [1]. MS principally alters the sensorimotor and cognitive functions. Its clinical development is highly unpredictable and is characterized by episodes of exacerbation (outbreaks) and subsequent deterioration. As a result, it produces progressive and permanent chronic disability for which there is currently no definitive treatment. Treating the disease, however, does improve its symptomatology and can modify its clinical course, making early diagnosis beneficial. Currently, MS diagnosis is performed using the McDonald criteria, which were recently revised [2]. Nonetheless, there is no definitive method of diagnosis. Studies to validate use of structural (magnetic resonance imaging (MRI) or optical coherence tomography (OCT)) and functional (visual evoked potentials (VEP), electroretinography (ERG), etc.) tests to support MS diagnosis are considered a high priority.

As the retina forms part of the CNS [3], MS’s neurodegenerative processes frequently manifest in the visual pathways [4], making analysis of them a means of diagnosing the disease and evaluating its advance. Alterations with axonal degeneration of the optic nerve are detected without and after episodes of optic neuritis (ON) (inflammation of the optic nerve), with affectation being greater in the latter case [5].

Using electrophysiological tests, it is possible to evaluate cell function by analyzing the electrical responses that neurons generate during synaptic transmission of information. In the case of MS, this altered function can be detected and quantified by tests such as measurement of visual evoked potentials, which evaluate the integrity of the posterior visual pathway [6], or by full-field electroretinography (ffERG) [7], which evaluates total retinal response [8].

Multifocal electroretinography (mfERG) was developed with the aim of obtaining both objective information on the cone-driven retina and a high-topographic-resolution sensitivity map of the retina [9,10]. This technique divides the retina into hexagonal regions or sectors scaled with eccentricity (typical values are 61 or 103 sectors, according to the ISCEV (International Society for Clinical Electrophysiology of Vision) standard for clinical multifocal electroretinography [11]). It takes a quasi-simultaneous recording of each of sector using a visual stimulation pattern governed by a pseudo-random white/black sequence (m-sequence) [12].

Stimulating 103 sectors obtains higher spatial resolution, while stimulating 61 sectors shortens the length of the test, and so obtains a better signal-to-noise ratio. The response most commonly used in mfERG is the first-order kernel, which generates a recording consisting of a negative wave (N1) followed by a maximum positive (P1), and occasionally, another negative wave (N2). N1 seems to represent the contribution from the cells that generate the a-wave, while P1 and N2 represent the cells generating the b-wave and oscillatory potentials in the light-adapted full-field ERG [11,12].

Traditional analysis of mfERG recordings is based on obtaining the amplitudes and latencies of these waves, and has been shown to be effective in diagnosing glaucoma [13] or age-related macular degeneration [14], among other diseases.

However, the few papers that have investigated the potential of mfERG analysis for diagnosing MS are based on traditional analysis of amplitudes and latencies and have produced conflicting findings [7,15,16,17,18,19,20], perhaps because of the heterogeneity of the subjects analyzed (different types of MS and presence/absence of ON) and the use of different mfERG technologies.

We hypothesize that the conflict in the results obtained to date with mfERG analysis can be resolved by analyzing recordings using advanced digital signal-processing algorithms. In line with this, research should be conducted into new filtering methods and into using signal features that allow practitioners to determine if the mfERG recordings are useful in MS diagnosis.

Filtering a signal consists of selecting certain components of that signal in order to improve their features. The most commonly used method is the Fourier transform, which breaks down a signal into a linear combination of sinusoidal functions and is the basis for the implementation of filters (analogue and digital). Other alternatives, such as the wavelet transform or empirical mode decomposition (EMD), open up the possibility of filtering a signal on the basis of features not contemplated by the Fourier method.

EMD allows time-frequency analysis of non-linear and non-stationary signals by decomposing them into a linear combination of oscillating components called intrinsic mode functions (IMFs), along with a residue [21]. EMD and its subsequent enhancements have been successfully applied in denoising signals [22,23], trend removal [24] and trend prediction [25], feature extraction in images [26], and pattern recognition [27].

If EMD is applied to signals that present intermittency or involve components with spectral proximity, the mode-mixing problem may arise (i.e., oscillations of very dissimilar amplitudes in a mode, or very similar oscillations in different modes). This drawback can be overcome by using ensemble empirical mode decomposition (EEMD) and adding white noise to the signal to be decomposed [28]. EEMD may also present the edge effect problem when cubic spline interpolation is used to fit the upper and lower envelopes of signals. A solution may be to obtain optimum endpoint values by minimizing the deviation evaluation function of the signal and signal envelope [29].

The original univariate EMD method has been extended to bivariate [30], trivariate [31], and multivariate signals [32] (multivariate EMD, MEMD). With multivariate signals, it is necessary to redefine the local maxima and minima and oscillatory modes in relation to the criteria applied to univariate signals. MEMD obtains projections that are evenly distributed in a multidimensional space. The multiple envelopes obtained are averaged and their extrema are then interpolated (using a cubic spline) to estimate the local n-dimensional mean. Each channel is given a specific threshold when selecting the relevant IMFs. In [22], the dependent thresholds are computed based on characteristics of white Gaussian noise and are applied to each channel separately; more recently, the Mahalanobis distance measure at multiple data scales obtained from multivariate empirical mode decomposition is proposed as a denoising method [33].

In biomedical signal analysis, EMD has been used to filter multifocal visual-evoked potentials in MS diagnosis (selecting only the IMF with the highest amplitude [34]) or to analyze gamma-band activity in single-channel electroencephalography (EEG) signals [35]. EEMD was used by Naik et al. [36] in single-channel electromyography (EMG) signal classification and by Chang to remove artefacts in electrocardiograms [37]. Multivariate signal analysis using MEMD has been employed in brain–computer interfaces [38], analysis of multichannel EEG signals [39], extraction of olfactory-induced hemodynamic responses in MRI blood oxygen level dependent (BOLD) signals [40], analysis of abdominal sounds [41], mechanomyography [42], and electrogastrography [43], among other applications.

To our knowledge, EMD has not been used in analysis of mfERG signals. The objective of this paper is to improve discriminant capacity in MS diagnosis by analyzing mfERG recordings using empirical mode decomposition filtering and employing the correlation between the signals and a normative database as an analysis parameter.

## 2. Materials and Methods

### 2.1. Clinical Study

The study protocol was approved by the Institutional Review Board of Miguel Servet University Hospital at the University of Zaragoza and adhered to the tenets of the Declaration of Helsinki. All participants provided informed consent.

The recordings were taken by the Ophthalmology Service at Miguel Servet University Hospital (Zaragoza, Spain). The signals were obtained from patients diagnosed with early-stage MS (in all cases < 12 months) with no history of optic neuritis, and from control subjects.

The participants had no concomitant ocular diseases, nor any previous history of retinal pathology, glaucoma, amblyopia, or significant refractive errors (more than 5 dioptres of spherical equivalent refraction or 3 dioptres of astigmatism), or systemic conditions that could affect the visual system.

A complete neuro-ophthalmic examination, including assessment of best-corrected visual acuity using the Snellen chart, contrast sensitivity using the CSV1000 test, color vision using Ishihara, pupillary reflexes, ocular motility, examination of the anterior segment, intraocular pressure (IOP) using the Goldmann applanation tonometer, and papillary morphology by fundoscopic exam, was performed on all subjects in order to detect ocular alterations (such as primary open-angle glaucoma, cataract or corneal pathology) that could affect functional vision or mfERG results.

### 2.2. Acquisition of mfERG Recordings

MfERGs were recorded according to the ISCEV standard [11], using a RETI-port/scan21 (Roland-Consult, Germany) device. The stimulus consisted of a set of 61 sectors, covering approximately 50 degrees in diameter of the central region of the visual field. A 21 inch monochrome CRT (cathode ray tube) monitor with a 60 Hz frame rate was used. The pupils were dilated and the eyes were adapted at room illumination prior to recording. A Dawson–Trick–Litzkow (DTL) electrode was placed on the lower eyelid conjunctiva after administering a topical anesthetic. An amplifier with a gain of 10^4^ and a bandwidth of 10–200 Hz was used. The sampling frequency was 1017 Hz and signal length was approximately 83 ms (n = 84 samples). Monocular recordings of both eyes were taken, randomly selecting the order of the tests for each subject. The records for the right eye of each participant were analyzed.

### 2.3. Empirical Mode Decomposition-based Filter

In this paper, we adopt the original EMD method given in Huang et al. [21], decomposing signal x(n) in a linear combination of L oscillating IMFs, plus a residue according to Equation (1):(1)$$x\left(n\right)=\overset{L}{\underset{k=1}{\sum }}{\mathrm{IMF}}_{k}\left(n\right)+{\mathrm{residue}}_{L}\left(n\right)$$

The first IMFs (k = 1, 2, …) represent the high-frequency components of x(n), while the higher IMFs (k = L, L−1, …) show the low-frequency information. Figure 1 shows an example of an mfERG signal decomposed into 4 IMFs (IMF_1_ … IMF_4_).

### 2.4. Analysis of mfERG Recordings

Briefly, the flow is shown in Figure 2:
(a)A template or normative database obtained from control subjects is computed for each sector by averaging those at the same location ($X_{S}^{\mathrm{TEM}}\left(n\right))$;(b)Each sector of each subject (control subjects and patients) is filtered using an adaptive EMD filtering block;(c)Sectors from the original, template, and filtered signals are grouped into 10 different clusters: whole visual field (SUM), 5 rings, and 4 quadrants;(d)As the analysis parameter, the Pearson correlation coefficient (PCC) is computed between each cluster and the corresponding template;(e)The area under the receiver operating characteristic curve (the mean area under the curve, AUC) is computed (control subjects versus patients).

#### 2.4.1. Calculation of the Normative Database

As set out in the ISCEV standard [11], a template was formed from age-similar control data obtained from the same laboratory. A normative database was built for the original signals (RAW) and for each of the sectors (S = 1, …, 61), averaging the traces of that sector with the sectors of the 6 control subjects (Equation (2)).
(2)$$X_{S}^{\mathrm{TEM}}\left(n\right)=\frac{1}{6}\overset{6}{\underset{i=1}{\sum }}x\left(n\right);\;S=1\ldots 61;n=1\ldots 84;$$

To avoid bias, when analyzing the eye of a control subject, its signals were not included in the template. Mathematically (Equation (3)), the template used to evaluate control subject j (j = 1, …, 6) was as follows:(3)$$X_{S}^{\mathrm{TEM},j}\left(n\right)=\frac{1}{6-1}\overset{6}{\underset{i=1}{\sum }}x\left(n\right);\;i\neq j,S=1\ldots 61;n=1\ldots 84;$$

#### 2.4.2. Adaptive EMD Filter

The adaptive filter is built independently for each of the 61 sectors of the visual field. For a recording $x_{S}^{\mathrm{RAW}}\left(n\right)$, the first 4 IMFs plus the corresponding residue are obtained, as per the algorithm described in [21], obtaining the following 4 approximations of the original signal (Equation (4)):(4)$$x_{S}^{k}\left(n\right)=\overset{4}{\underset{k}{\sum }}{\mathrm{IMF}}_{k}\left(n\right)+{\mathrm{residue}}_{L=4}\left(n\right);\;k=1\ldots 4;$$

Examination of the control database confirmed that the first 4 IMFs (L = 4: IMF_1_ + IMF_2_ + IMF_3_ + IMF_4_) contain 92.23% of the energy of the original mfERG signals. Equation (4) produces 4 approximations of signal x(n), in which the highest frequency components (IMF1, IMF2, IMF3) are progressively eliminated.

Of the 4 approximations obtained with Equation (4), the one containing the useful information is selected according to the principle of partial reconstruction. There are several options, such as energy-based methods [44], correlation-based methods [45,46,47], probability density function based methods [48], entropy [49], higher order statistics [50], mutual information [51], and mutual information entropy [52]. In this paper, the method described in [47] has been used, selecting as the filtered signal the one among the 4 approximations obtained in Equation (4) that has the highest Pearson correlation coefficient with the signal from the same sector of the normative database. Therefore, the filtered signal corresponding to sector S (S = 1, …, 61) is (Equation (5)):(5)$$X_{S}^{\mathrm{EMD}}\left(n\right)=x_{S}^{k}\left(n\right)|\mathrm{maxPearson}\_\mathrm{corr}\left(X_{S}^{\mathrm{TEM}}\left(n\right),x_{S}^{k}\left(n\right)\right);\;k\;=\;1,\;\ldots ,\;4;$$

If in Equation (5) all the correlations are negative, then the sector is considered non-analyzable (NaS); this can occur in records of sectors that are heavily contaminated by noise.

#### 2.4.3. Cluster Calculation

The traces of the 61 mfERG sectors in all three cases (RAW signals, templates, and filtered signals) are averaged to form 10 different clusters: 5 rings (R1, …, R5), 4 quadrants (IN: inferior nasal quadrant; SN: superior nasal quadrant; ST: superior temporal quadrant; IT: inferior temporal quadrant), and an average of the 61 sectors (SUM), according to the ISCEV standard [11]. The signal clusters are named as follows: $X_{\mathrm{Ring}1}\left(n\right),\ldots X_{\mathrm{Ring}5}\left(n\right),X_{\mathrm{ST}}\left(n\right),\ldots X_{\mathrm{SUM}}\left(n\right)$. Figure 3 shows the numbering assigned to each sector, as well as the clusters analyzed.

MfERG tests generate a large number of single responses in different sectors of the retina. However, considering each single sector separately by itself does not have value for diagnosis because its signal-to-noise ratio is very low. To overcome this drawback, a signal level fusion model from multiple locations of the retina has been implemented by using a clustering in rings, quadrants, and the whole visual field. The goal of this data fusion approach is to obtain valuable diagnostic information.

#### 2.4.4. Analysis Parameters

In order to compare our results with the classic method of analyzing amplitudes and latencies, in this paper we use the amplitude of wave N1 (A^N1^) measured over the original signals, since in our database it is the variable that exhibits the best capacity to discriminate between control subjects and patients. The location of wave N1 corresponds to the minimum recording inside the 9–32 ms temporal window and the N1 amplitude is the difference between baseline and N1 in terms of absolute value.

As a new feature of the signals, the Pearson correlation coefficient (PCC) is obtained to discriminate between signals from control subjects and signals from MS patients. As an example, if Ring 2 is analyzed, the discriminant parameter for the RAW signal is defined as:(6)$${\mathrm{PCC}}_{\mathrm{Ring}2}^{\mathrm{RAW}}=\mathrm{Pearson}\_\mathrm{corr}\left(X_{\mathrm{Ring}2}^{\mathrm{RAW}}\left(n\right),X_{\mathrm{Ring}2}^{\mathrm{TEM}}(n\right));$$

And for the signal filtered using EMD:(7)$${\mathrm{PCC}}_{\mathrm{Ring}2}^{\mathrm{EMD}}=\mathrm{Pearson}\_\mathrm{corr}\left(X_{\mathrm{Ring}2}^{\mathrm{EMD}}\left(n\right),X_{\mathrm{Ring}2}^{\mathrm{TEM}}(n\right));$$

### 2.5. Statistical Analysis

The results were expressed as the mean and the standard deviation. All statistical analyses were performed using the SPSS (Statistical Package for Social Sciences) 25.0 software (SPSS Inc. Chicago, Illinois, USA). A p value below 0.05 was considered statistically significant.

The normality of the results was assessed using the Shapiro–Wilk test. The differences between groups were evaluated using the dependent t-test (paired-samples t-test) in normal distributions or the Mann–Whitney U test in non-normal distributions.

The AUC [53] was employed to assess the discrimination capability of each of the features analyzed in this study.

## 3. Results

Right-eye mfERG recordings from fifteen subjects (age = 44.46 ± 8.24 years; male/female = 3:12) diagnosed with early-stage MS with no history of optic neuritis were used. Recordings were also taken from six control subjects (age = 35.83 ± 10.65 years; male/female = 3:3). There was no significant difference between patient and control subject ages (t-test, p = 0.060).

### 3.1. Normative Database

Figure 4a shows the template obtained for the 61 sectors by averaging the RAW signals from the 6 control subjects used to filter the MS patient group: $X_{S}^{\mathrm{TEM}}$ (Equation (2)).

### 3.2. Adaptive EMD Filter

Table 1 shows the number of IMFs used in adaptive filtering. In most cases, filtering is done using the sum of IMFs 2, 3, and 4 for both the control subjects (57.38%) and the patients (41.31%). In the case of the control subjects, 0.82% of the signals are non-analyzable, a figure that rises to 10.49% in MS patients.

Figure 4b shows the original and filtered signals for a patient. In this particular case, two sectors are not analyzable (2 and 14) and in two other sectors the filtered signal matches the original (k = 1, no IMFs have been discarded), namely sectors 4 and 21.

### 3.3. Analysis of Discriminant Capacity

Among the traditional methods for analyzing latencies and amplitudes, the best option (i.e., the one that produces the highest AUC values) takes the amplitude of wave N1 as a parameter. For purposes of comparison, Table 2 shows the results obtained using 3 analysis methods: (a) analysis of the amplitude of wave N1 (AN1); (b) correlation coefficient between the template and the RAW signals (unfiltered); and (c) correlation coefficient between the template and the mfERG signals filtered using the adaptive EMD method. The values obtained in each cluster (SUM, …, IT) and the p and AUC values between control subjects and patients are included.

In conventional analysis of the amplitude of wave N1, significant differences between patients and control subjects are only found in ring 3 and the inferior nasal (IN) quadrant (p = 0.023, p = 0.036, respectively, Mann–Whitney U test), producing AUC values of 0.82 and 0.80, respectively. The average AUC value for all the clusters analyzed using the amplitude of N1 is AUC = 0.63.

Using the value of the correlation between the signals and the normative database as the analysis parameter improves discriminant capacity. If this process is performed using unfiltered (RAW) signals, the maximum AUC value is obtained in ring 4 (AUC = 0.89) and the mean value for all clusters is AUC = 0.83. As expected, the value of the correlation coefficient is higher in the control subjects than in the patients, as since there is no dysfunction in the retina, the response is less variable, and therefore, more similar to the normative database.

The difference between control subjects and MS patients is accentuated by combining EMD filtering of the recordings with subsequent calculation of correlation with the normative database. In this case, there is significant difference in all the clusters considered (p < 0.014) and minimum AUC = 0.83 (ring 1), producing a mean value for all the clusters analyzed of 0.90. The greatest discriminant capacity is obtained in ring 4 (AUC = 0.96) and in the IN quadrant (AUC = 0.94). Figure 5 shows the discriminant capacity of the methods tested for the different clusters.

## 4. Discussion

The purpose of this paper has been to research new methods of processing mfERG recordings to increase MS diagnosis capacity. The few papers that analyze the applicability of using mfERG recording analysis to diagnose MS are based on analysis of the amplitudes and latencies of first-order kernel waves, and their findings present discrepancies. These different findings may be partly due to the fact that electrophysiological tests are subject to the variability introduced by technical factors, such as device type, hardware configuration, electrodes, positioning, flash features, test with different number of stimuli arrays, or clusters analyzed (rings or quadrants). Physiological factors—such as pupil dilation or even temperature and oxygenation—can also introduce variability [54]. It should likewise be noted that the studies conducted include patients with different subtypes of the disease and with or without optic neuritis (see Table 3).

In MS diagnosis based on the amplitude and latency parameters of first-order kernel waves in previous studies, prolonged latency is the altered parameter found most frequently [7,15,16,18,19], and even Neroev et al. [19] suggested using the latency of wave P1 in the parafovea as a marker of MS progression. The studies that detected the classical parameters with greatest alteration (lower amplitudes or higher latencies) in MS patients generally employed a 61-hexagon test [7,16,18,19].

The studies that employed 103 hexagons either did not find any statistically significant differences (SSDs) between MS patients and healthy subjects [15] or found normal latencies [17]. In our case, the 61-hexagon test was selected because it was shorter. This was to avoid fatigue, and with it, possible fixation losses, noise, signal quality issues, and errors due to tiredness, and so avoid retests that could influence the results [55]. The strength of our results—which did show SSDs even when analyzing a small number of subjects—supported by existing evidence, seem to suggest that in MS patients, conducting short tests with 61 hexagons is more appropriate.

Another aspect that may explain why there are no clear findings that support the ability of mfERGs to diagnose MS may be the need to use advanced processing algorithms and to develop parameters with greater diagnostic capacity. This paper proposes using an EMD-based method to filter mfERG recordings and employing the correlation of filtered signals with a normative database as the analysis parameter. The findings of this paper show a clear potential marker ($\mathrm{mean}({\mathrm{AUC}}^{\mathrm{FILTER}}=0.90$) of the alterations in the mfERG responses in the early stages of MS, which if confirmed in a broader study, could enable clinical applicability.

When looking for a deeper justification for the findings, the involvement of glutamate in MS and the origin of the mfERG recordings should be analyzed. Glutamate is the central nervous system’s main excitatory neurotransmitter, but when present in excess it triggers a chain of negative reactions. Glutamate neurotoxicity’s involvement in the pathogenesis of demyelination and neuronal and synaptic damage in MS has been demonstrated on numerous occasions [56]; activated immune cells and astrocytes release large amounts of glutamate, damaging myelin sheaths and axons [57,58], and changes in glutamate levels in these areas have been associated with the later stages of MS, episodes, and secondary progression. An imbalance in glutamate receptor levels and expression can also occur in earlier stages or even when no damage is observed in white matter with MRI. High levels of glutamate have been found in serum and white and grey matter [59,60]. On the other hand, neurodegenerative diseases, such as MS, may not affect all cell types equally, making studying types rather than individual neurons more appropriate.

The mfERG wave is the overlapping of the responses of ON (activated) OFF (desactivated) bipolar cells, with a small contribution made by the inner retina and photoreceptors (mainly cones) [61]. In total, 12 types of bipolar cells have been described. These are progressively stratified in depth in the inner nuclear layer of the retina and form synaptic connections with the ganglion cells in the inner plexiform layer and with the photoreceptors in the outer plexiform layer. Studies on animals have shown that most bipolar cells appear to have connections with cones. These are sometimes specific between particular bipolar cell and cone types (e.g., M cone and CBC1, or S cone and CBC9), and only one type of bipolar cell would connect with the rods, although there also appear to be connections between rod and cone pathways [62]. Bipolar cells respond to the release of glutamate by photoreceptors in their synaptic communication [63] and adopt a mosaic arrangement, in which the overlapping contacts are potential synaptic pairings. Connectivity can be studied using electron microscopy, molecular markers, or more recently, computational methods. However, in all these cases histological tissue is required [64]. Using mfERG allows practitioners to study the mediated response of these synaptic responses non-invasively in vivo.

In animal studies using electrophysiological analysis, Vielma and Schmachtenberg [63] reported the existence of various types of OFF bipolar cells that exhibit different responses depending on glutamate levels; specifically, bipolar cells BC 3a, 3b, and 4 exhibited a significant inhibitory input in response to glutamate and were subject to inhibitory modulation with nitric oxide (NO). They also showed 6 functionally distinguishable types of bipolar cells with combinations of individual glutamate channels and receptors that give them unique electrophysiological filtering and signal-processing properties.

In justifying the findings, it can be hypothesized that using advanced filtering techniques such as EMD has the capacity to separate the responses of different bipolar cell types.

If our hypothesis can be corroborated, the features of the waves could be topographically extrapolated to photoreceptor/bipolar ON/bipolar OFF synaptic functionality. Although most studies detected increased P1 affectation, in our case it was the amplitude of N1 that showed the greatest affectation and diagnostic reliability. This may be a reflection of an early photoreceptor/bipolar cell synaptic alteration due to glutamate disbalance in MS. In addition, the inferior nasal quadrant showed the greatest discriminant capacity, which would correspond topographically to lower temporal affectation of the optic disc, which is typically affected in this disease.

Our findings show that as in previous studies [7,16], the outermost rings (R4 in our case)—as well as the nasal quadrant [18]—and therefore those close to the optic disc, show greater alteration. In the case of [18], significant affectation in mfERG amplitude and latency at the inferonasal (p = 0.045) and superonasal (p = 0.042) quadrants is reported in MS patients with and without prior ON. In our study, the decision to analyze only MS without ON, and therefore in the earliest stage, may be the reason why affectation was not (yet) found in the superonasal quadrant.

Our results showed higher N1 amplitudes (cone response reflex) in MS patients than in healthy control subjects. This may be due to the cones adapting to light, as described by Harrison et al. in 2018 [55], who detected greater (not clinically relevant) amplitudes in the mfERG recordings after exploration with ffERG. As these authors found, the effect is not very great, which is why our findings do not show statistically significant differences. Repetition of the exploratory phases in MS patients due to increased tiredness compared to healthy control subjects was likely to produce this long-term adaptation of the cones, as was mentioned earlier. Another possible explanation would concur with the results obtained in other neurodegenerative diseases such as Alzheimer’s; Lopez et al. [65] found increased functional brain activity in anterior circuits using magnetoencephalography in mild cognitive impairment (an early condition in dementia). In [66], it was suggested that this hyperactivity may be one of the earliest dysfunctions of the pathophysiological process of neurodegeneration and would subsequently decline as dementia develops [67].

Our hypothesis is that the glutamate imbalance would be transported anterogradely from the brain by the axons, reaching the optic nerve. It would be possible to detect topographical retinal dysfunction in the outer rings and the nasal quadrant in the form of an increase in mfERG amplitude in early-stage MS. With time, in later or more severe stages [7,15,17,18,19] there would be a decline, as was detected in [17], in which 5 out of 7 patients showed decreased amplitude with predominant macular thinning when examined using optical coherence tomography.

In conclusion, a minimally invasive, simple, and accessible test such as mfERG could detect a glutamate imbalance in MS, even in very early stages of onset or without MRI evidence of damage, which would facilitate topographical diagnosis. If our hypothesis were to be corroborated, it would even allow practitioners to view the advance of secondary degeneration and the efficacy of glutamatergic balance-regulating treatments over time [68].

### Limitations and Future Work

The main limitation of our study is the small size of our database, although the results obtained are statistically significant and a clear potential biomarker has been identified.

We also consider it necessary to conduct longitudinal follow-up and to monitor late stages of disease onset to ascertain how this response evolves, as it could indicate the topography of secondary neurodegeneration or dissemination or those areas most susceptible to damage. It would also be interesting to assess it during symptomatic episodes.

On the other hand, it would be interesting to analyze and compare the mfERG responses in this same type of patient but using larger exploratory sectors, as well as adopting shorter exploratory protocols to make it a faster, simpler, and more comfortable test for the patient, and so reduce the need for re-explorations that could influence the results due to light adaptation.

In this study, glutamate levels have not been analyzed, and most patients were being treated with anti-CD-20 (fingolimod, adalimumab), which could alter these levels due to immunity control.

Regarding our second hypothesis that decomposition of mfERG waves could reveal the responses of different bipolar cell types; as described above, this could open the way for both basic or translational and clinical studies.

From the point of view of signal analysis, since the basic version of the EMD algorithm has been used in this paper to analyze univariate signals, it would be worth evaluating the merits of multivariate decomposition (MEMD).

The latest clinical decision support systems for diagnosing MS are based on artificial intelligence analyze multifocal visual evoked potentials (mfVEP) [34], optical coherence tomography [69], EEG signals [70], functional magnetic resonance imaging (fMRI) [71], among others. Biomedical data fusion with advanced strategies for the analysis by selecting those more discriminant parameters from each exploratory test would reduce the decision error probability, increase reliability, and therefore reach an earlier and more precise diagnosis.

## 5. Conclusions

This paper has researched a possible method of obtaining a valid MS biomarker by analyzing mfERG recordings. Previous papers have analyzed the morphological characteristics of the signals (wave amplitudes and latencies), leading to conflicting results in different studies. By combining signal filtering with EMD and by using the correlation with signals from a normative database as an analysis parameter, a clear marker (mean value of AUC = 0.90) has been obtained, having greater discriminant capacity in ring 4 (AUC = 0.96) and in the lower nasal quadrant (AUC = 0.94). Although these results cannot be generalized due to the small data size and need to be confirmed in a larger study, they suggest that the combination of filtering mfERG recordings using EMD and calculating the correlation with a normative database would make mfERG waveform analysis applicable to assessment of multiple sclerosis in early stage patients.

## Figures and Tables

**Figure 1 sensors-20-00007-f001:**
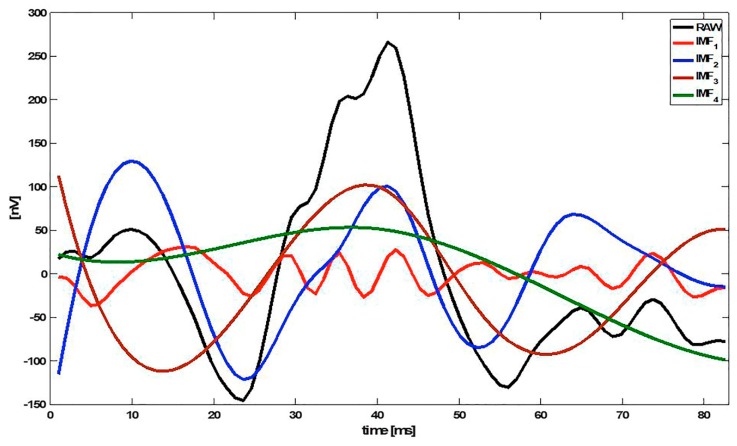
Example of an multifocal electroretinogram (mfERG) signal decomposed into 4 intrinsic mode functions (IMFs). Note: RAW = original signal.

**Figure 2 sensors-20-00007-f002:**
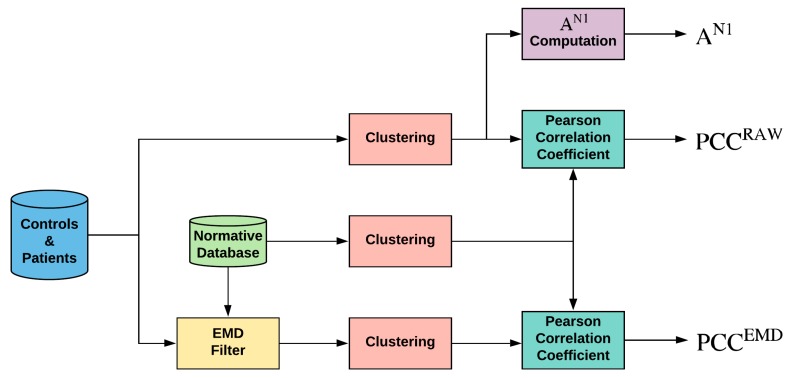
General diagram of the work performed. Note: EMD = empirical mode decomposition; PCC = Pearson correlation coefficient; A^N1^ = amplitude of wave N1 of original signals.

**Figure 3 sensors-20-00007-f003:**
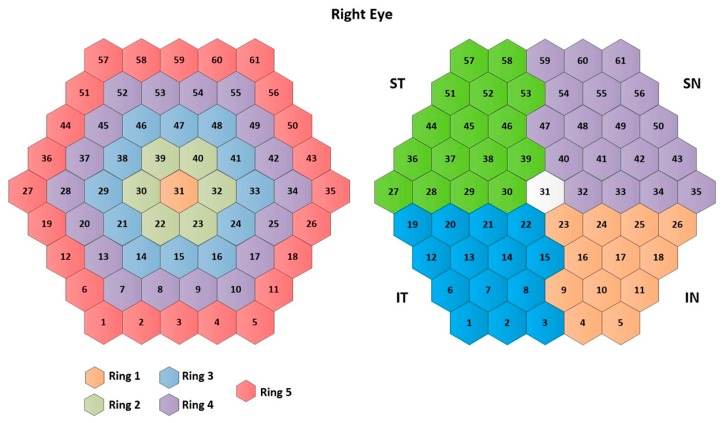
Cluster definition. Note: IN= inferior nasal quadrant; SN= superior nasal quadrant; ST= superior temporal quadrant; IT= inferior temporal quadrant.

**Figure 4 sensors-20-00007-f004:**
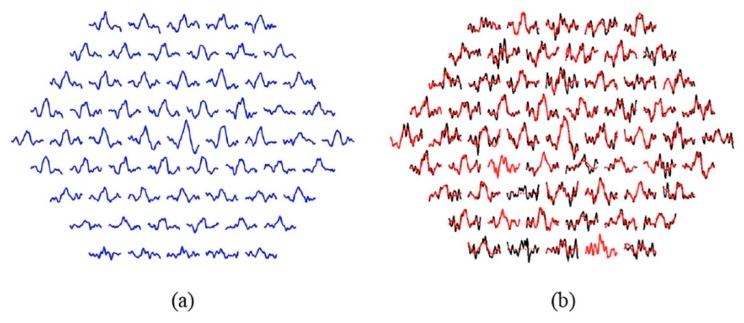
(**a**) Normative database according to Equation (2). (**b**) Effect of EMD filtering. Note: Black traces= original signals; red traces= filtered signals. Sectors showing a dotted red line (2 and 14): the sector is not analyzable. Sectors showing only red (4 and 21): the filtered signal is equal to the original signal.

**Figure 5 sensors-20-00007-f005:**
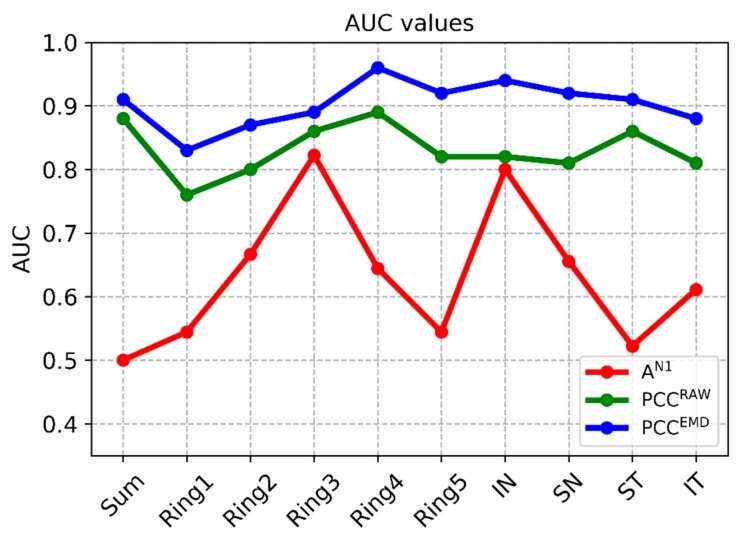
AUC values in the different methods. Note: PCC^EMD^= Pearson correlation coefficient of filtered signals; PCC^RAW^= Pearson correlation coefficient of original signals; A^N1^ Amplitude= amplitude of wave N1 of original signals; SUM= whole visual field; IN= inferior nasal quadrant; SN= superior nasal quadrant; ST= superior temporal quadrant; IT= inferior temporal quadrant.

**Table 1 sensors-20-00007-t001:** Best intrinsic mode functions (IMFs).

k Value Selected in Equation (5)	IMFs Used	Control Subjects	Multiple Sclerosis (MS) Patients
k = 1	IMF1, IMF2, IMF3, IMF4 + residue	23.50%	9.84%
k = 2	IMF2, IMF3, IMF4 + residue	57.38%	41.31%
k = 3	IMF3, IMF4 + residue	16.39%	28.31%
k = 4	IMF4 + residue	1.91%	10.05%
NaS		0.82%	10.49%

**Table 2 sensors-20-00007-t002:** Results achieved using the implemented methods.

	Amplitude Analysis		Correlation Analysis: PCC			
	A^N1^ (nV) (mean ± SD)		RAW Signals (mean ± SD)		EMD Filtered Signals (mean ± SD)	
	Controls	MS Patients	Controls	MS Patients	Controls	MS Patients
**Sum**	160.41 ± 53.87	155.44 ± 82.84	0.96 ± 0.03	0.73 ± 0.32	0.98±0.02	0.91 ± 0.06
	p = 1 ^#^, AUC = 0.50		p = 0.05 ^#^, AUC = 0.88		p = 0.002 ^#^, AUC = 0.91	
**Ring1**	308.17 ± 79.95	533.49 ± 456.84	0.90 ± 0.04	0.56 ± 0.42	0.95 ± 0.03	0.61 ± 0.39
	p = 0.79 ^#,^ AUC = 0.54		p = 0.095 ^#^, AUC = 0.76		p = 0.014 ^#^, AUC = 0.83	
**Ring2**	207.73 ± 59.18	274.29 ± 130.37	0.93 ± 0.04	0.71 ± 0.29	0.96 ± 0.03	0.83 ± 0.17
	p = 0.25 *, AUC = 0.67		p = 0.055 ^#^, AUC = 0.80		p = 0.006 ^#^, AUC = 0.87	
**Ring3**	179.97 ± 45.95	266.06 ± 96.70	0.93 ± 0.04	0.65 ± 0.40	0.96 ± 0.03	0.82 ± 0.17
	p = 0.023 ^#^, AUC = 0.82		p = 0.006 ^#^, AUC = 0.86		p = 0.005 ^#^, AUC = 0.89	
**Ring4**	161.17 ± 58.22	201.03 ± 92.54	0.91 ± 0.05	0.63 ± 0.35	0.95 ± 0.02	0.80 ± 0.15
	p = 0.34 **^#^**, AUC = 0.64		p = 0.005 ^#^, AUC = 0.89		p = 0.001 ^#^, AUC = 0.96	
**Ring5**	158.06 ± 63.45	176.27 ± 90.63	0.90 ± 0.06	0.62 ± 0.34	0.95 ± 0.04	0.82 ± 0.11
	p = 0.79 **^#^**, AUC = 0.54		p = 0.023 ^#^, AUC = 0.82		p = 0.002 ^#^, AUC = 0.92	
**IN**	137.41 ± 66.43	258.50 ± 136.53	0.92 ± 0.04	0.69 ± 0.29	0.97 ± 0.02	0.86 ± 0.14
	p = 0.036 ^#^, AUC = 0.80		p = 0.018 ^#^, AUC = 0.82		p = 0.002 ^#^, AUC = 0.94	
**SN**	188.45±70.19	227.41±88.25	0.92 ± 0.04	0.59±0.46	0.96 ± 0.01	0.78 ± 0.19
	p = 0.30 **^#^**, AUC = 0.66		p = 0.018 ^#^, AUC = 0.81		p = 0.002 ^#^, AUC = 0.92	
**ST**	195.55 ± 41.52	228.98 ± 200.02	0.93 ± 0.04	0.70 ± 0.32	0.96 ± 0.02	0.83 ± 0.13
	p = 0.70, AUC = 0.52		p = 0.011 ^#^, AUC = 0.86		p = 0.001 ^#^, AUC = 0.91	
**IT**	148.12 ± 67.56	186.46 ± 95.40	0.87 ± 0.05	0.60 ± 0.34	0.93 ± 0.04	0.75 ± 0.19
	p = 0.42 ^#^, AUC = 0.61		p = 0.045 ^#^, AUC = 0.81		p = 0.005 ^#^, AUC = 0.88	
**Mean AUC**	$\mathrm{mean}\left({\mathrm{AUC}}^{N1}\right)=0.63$		$\mathrm{mean}\left({\mathrm{AUC}}^{\mathrm{PCC}\_\mathrm{RAW}}\right)=0.83$		$\mathrm{mean}\left({\mathrm{AUC}}^{\mathrm{PCC}\_\mathrm{EMD}}\right)=0.90$	

Note: * t-test; ^#^ Mann–Whitney U test; bold = significance difference. AUC = mean area under the curve; A^N1^= amplitude of wave N1 of original signals.

**Table 3 sensors-20-00007-t003:** Studies analyzing mfERG amplitude and latency in multiple sclerosis (MS) diagnosis.

Authors	Commercial mfERG System	Hexagons	Rings/Quadrants	MS	Results
Moura et al. (2007) [15]	VERIS Electro-Diagnostic-Imaging	103	Rings	MS with + without ON	Decreased amplitude Prolonged latency (p > 0.05)
Saidha et al. (2011) [17]	VERIS Electro-Diagnostic-Imaging	103	Rings	MS with + without ON (RR, RP, SP)	Decreased amplitude Normal latency
Neroev et al. (2016) [19]	RetiPort Roland-Consult	61	Rings	MS with ON	Decreased amplitude Prolonged latency
Vildades et al. (2017) [18]	RetiPort Roland-Consult	61	Quadrants	MS with + without ON	Decreased amplitude Prolonged latency
Gundogan et al. (2007) [16]	RetiScan Roland-Consult	61	Rings	MS without ON	Tendency to prolonged latency (p > 0.05)
Hanson et al. (2018) [7]	Espion Diagnosis LLC	61	Rings	MS with + without ON (RR, PP)	Prolonged latency

Note: RR= relapsing-remitting; PP= primary-progressive; SP= secondary-progressive; ON= optic neuritis.

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
