# Peer review of "Empirical Mode Decomposition-Based Filter Applied to Multifocal Electroretinograms in Multiple Sclerosis Diagnosis"

_sensors, 2019, doi:10.3390/s20010007_

Round 1

Reviewer 1 Report

The work is interesting and has a large application potential, however, the research was conducted on a small number of patients. This means that the results cannot be generalized too much. I suggest that the authors expand this part of the work showing the pros and cons of this method. In addition, my other comments:

line 47, Why the number of sectors is 61 and 103? what's the reason?
line 57-64, no other methods are available to diagnose MS. It is difficult to refer to only one method. I suggest expanding this paragraph.

line 89-96, why such measurement parameters were used? what was it dictated? Has any optimization been carried out?

Figure 2, this figure needs to be corrected because the descriptions are ambiguous
line 223-239, why for the determined values no drawing was made for better reading?

Reviewer 2 Report

This manuscript reports research on Empirical mode decomposition-based filter applied to multifocal electroretinograms in multiple sclerosis diagnosis. The research is interesting however, authors need to address some of the major concerns before the paper is considered for publication.

There are several recent/past studies used EMD, EEMD and LMD for various signal processing methods including feature extraction, classification etc. Also, there are lots of research which used non-linear signal analysis and phase construction as well. These papers needs to be studied and cited in the revised manuscript. Please cite the following papers related to the field and its applications in the revised version.

Guo, Yina, et al. "Edge effect elimination in single-mixture blind source separation." Circuits, Systems, and Signal Processing 32.5 (2013): 2317-2334.

Hao, Huan, H. L. Wang, and N. U. Rehman. "A joint framework for multivariate signal denoising using multivariate empirical mode decomposition." Signal Processing 135 (2017): 263-273.

Naik, Ganesh R., S. Easter Selvan, and Hung T. Nguyen. "Single-channel EMG classification with ensemble-empirical-mode-decomposition-based ICA for diagnosing neuromuscular disorders." IEEE Transactions on Neural Systems and Rehabilitation Engineering 24.7 (2016): 734-743.

Chen, Yong, Chun-ting Wu, and Huan-lin Liu. "EMD self-adaptive selecting relevant modes algorithm for FBG spectrum signal." Optical Fiber Technology 36 (2017): 63-67.

Specific comments to each section

Introduction section need to be further improved with more explanation and literature review.

What criteria was used to select the number of IMFs from the data? This needs to be clearly explained in the revised manuscript.

The optimal selection of IMFs used for the proposed method needs to be clearly explained in the manuscript.

Please compare the proposed method with the other state of the methods available in the literature.

Please include the detailed statistical analysis for the results.

Discussion and conclusion section need to be further improved.

Round 2

Reviewer 1 Report

I think that the answers given to me are satisfying. In addition, it can be seen that the authors made significant changes to make the manuscript at the highest scientific level. I recommend it for further stages of evaluation.

Author Response

Thank you very much for recommending our manuscript for further stages of evaluation

Reviewer 2 Report

The authors have addressed all my comments satisfactorily and the paper can be considered for publication.

Author Response

thank you very much for considering our manuscript for further stages of publication.